# Therapeutic Effects of *Morinda citrifolia* Linn. (Noni) Aqueous Fruit Extract on the Glucose and Lipid Metabolism in High-Fat/High-Fructose-Fed Swiss Mice

**DOI:** 10.3390/nu12113439

**Published:** 2020-11-10

**Authors:** Aline Carla Inada, Gabriela Torres Silva, Laleska Pâmela Rodrigues da Silva, Flávio Macedo Alves, Wander Fernando de Oliveira Filiú, Marcel Arakaki Asato, Wilson Hino Kato Junior, Joaquim Corsino, Patrícia de Oliveira Figueiredo, Fernanda Rodrigues Garcez, Walmir Silva Garcez, Renée de Nazaré Oliveira da Silva, Rosangela Aparecida dos Santos-Eichler, Rita de Cássia Avellaneda Guimarães, Karine de Cássia Freitas, Priscila Aiko Hiane

**Affiliations:** 1Post Graduate Program in Health and Development in the Central-West Region of Brazil, Federal University of Mato Grosso do Sul, Campo Grande, MS 79070-900, Brazil; gabitorres483@gmail.com (G.T.S.); laleskaprodrigues@gmail.com (L.P.R.d.S.); rita.guimaraes@ufms.br (R.d.C.A.G.); kcfreitas@gmail.com (K.d.C.F.); priscila.hiane@ufms.br (P.A.H.); 2Institute of Biosciences, Federal University of Mato Grosso do Sul-UFMS, Campo Grande, MS 79070-900, Brazil; flaurace@yahoo.com.br; 3Faculty of Pharmaceutical Science, Food and Nutrition, Federal University of Mato Grosso do Sul-UFMS, Campo Grande, MS 79070-900, Brazil; wander.filiu@gmail.com; 4Faculty of Medicine, Federal University of Mato Grosso do Sul—UFMS, Campo Grande, MS 79070-900, Brazil; marcel_arakakiasato@hotmail.com; 5Laboratory PRONABio (Laboratory of Bioactive Natural Products)—Chemistry Institute, Federal University of Mato Grosso do Sul-UFMS, Campo Grande, MS 79070-900, Brazil; hinokato@gmail.com (W.H.K.J.); corsinojoaquim@gmail.com (J.C.); patriciadeoliveirafigueiredo@gmail.com (P.d.O.F.); fernandagarcez@gmail.com (F.R.G.); walmir.garcez@ufms.br (W.S.G.); 6Department of Pharmacology, Biomedical Sciences Institute, University of São Paulo, São Paulo, SP 05508-900, Brazil; oliveirarenee@gmail.com (R.d.N.O.d.S.); reichler@icb.usp.br (R.A.d.S.-E.)

**Keywords:** metabolic syndrome, synergistic effect, de novo lipogenesis

## Abstract

The aim of this study was to evaluate the therapeutic effects of two different doses (250 and 500 mg/kg) of *Morinda citrifolia* fruit aqueous extract (AE) in high-fat/high-fructose-fed Swiss mice. The food intake, body weight, serum biochemical, oral glucose tolerance test (OGTT), and enzyme-linked immunosorbent assay (ELISA), as well as histological analyses of the liver, pancreatic, and epididymal adipose tissue, were used to determine the biochemical and histological parameters. The chemical profile of the extract was determined by ultra-fast liquid chromatography–diode array detector–tandem mass spectrometry (UFLC–DAD–MS), and quantitative real-time PCR (qRT-PCR) was used to evaluate the gene expressions involved in the lipid and glucose metabolism, such as peroxisome proliferative-activated receptors-γ (PPAR-γ), -α (PPAR-α), fatty acid synthase (FAS), glucose-6-phosphatase (G6P), sterol regulatory binding protein-1c (SREBP-1c), carbohydrate-responsive element-binding protein (ChREBP), and fetuin-A. Seventeen compounds were tentatively identified, including iridoids, noniosides, and the flavonoid rutin. The higher dose of AE (AE 500 mg/kg) was demonstrated to improve the glucose tolerance; however, both doses did not have effects on the other metabolic and histological parameters. AE at 500 mg/kg downregulated the PPAR-γ, SREBP-1c, and fetuin-A mRNA in the liver and upregulated the PPAR-α mRNA in white adipose tissue, suggesting that the hypoglycemic effects could be associated with the expression of genes involved in de novo lipogenesis.

## 1. Introduction

Noncommunicable diseases (NCDs) are considered one of the major health risks of modern society and are increasing in morbidity and mortality in developed and undeveloped countries [1]. Metabolic syndrome (MetS) is among these NCDs [1], and, although MetS started in the Western world, it has become a global problem. MetS is not considered a single pathological condition, but a cluster of metabolic abnormalities, including abdominal obesity, insulin resistance (or type 2 diabetes mellitus), systemic hypertension, and atherogenic dyslipidemia [2] and the hepatic component, such as non-alcoholic fatty liver disease (NAFLD) [3]. The two main factors that may be responsible for spreading MetS are the adherence to the Western lifestyle that is characterized by the consumption of non-healthy food known as the Western-type diet, which is composed of high-calorie/low-fiber fast food, in association with a sedentary lifestyle [1].

High-fat and high-carbohydrate diets, such as the high-fat/high-fructose (HFF) diet, are examples of a Western-type diet and generally contain fructose and sucrose, as well as saturated fat [2]. Fructose, which is considered the sweetest of all naturally occurring carbohydrates, is found as a hexose in fruits and honey [4] and is used commercially in juice, soft drinks, high-fructose corn syrup, and baked goods [5]; therefore, the consumption of fructose has increased considerably in everyday diets [5]. The fructose that is present in fruits and honey is considered a modest component of energy intake by individuals; unfortunately, the vast majority of fructose in the diet is due to added sugars, which refers to sugars not naturally occurring in foods. The two most important sources of added sugars are sucrose (containing 50% of fructose) and high-fructose corn syrup (containing 42–55% fructose) [4].

Evidence demonstrated the use of aqueous extracts from *Phyllanthus emblica* fruits [6] and natural juices, such as orange and pomegranate [7,8], as potential nutraceuticals to prevent or treat metabolic disturbances associated with MetS. Another example of fruits with nutraceutical properties may be attributed to *Morinda citrifolia* Linn. (*M. citrifolia*), popularly known as noni fruit and Indian mulberry, used in the folk medicine of Polynesians 200 years ago [9]. In 1996, noni fruit juice began to be commercialized in the USA [10,11], and, after that, in 2003, the European Commission approved the commercialized fermented *M. citrifolia* fruit juice (Tahitian Noni Juice^®^) as a novel food [10].

Although, in the USA, Europe, and Asia, the commercialization of *M. citrifolia* products, such as juice and encapsulated powder, is common [12], in Brazil, this market is not yet legalized, due to controversies that revolve around consuming *M. citrifolia* fruit [13]. Certain studies reported hepatotoxicity signs [14,15], whereas other studies did not demonstrate toxic effects [16,17]. Another issue is regarding the lack of pharmacological studies that standardize doses, times of treatment, adverse effects, and interactions of products derived from noni fruit with other conventional drugs [13]. The astringent odor and bitter taste of *M. citrifolia* fruit led companies to produce manufactured juice with the addition of other types of fruit juices, including grape, blueberry, and cranberry, in order to mask the taste [12,18], leading to the consideration that the actions of bioactive compounds from *M. citrifolia* fruit could not be attributed solely to themselves due to the addition of other bioactive components from the other fruits.

The *M. citrifolia* fruit processing should also be considered relevant, as it can be manufactured in many ways, including the fermentation of fruit juice kept in sealed containers for 4–8 weeks, which is one of the most common processing methods [9,12]. Food processing may activate some beneficial microorganisms to promote a probiotic effect or deactivate spoilage microorganisms, and enzymes may modify certain nutritional factors or the phytochemical composition and organoleptic properties of the products [12,19,20]. Nutraceutical products are functional foods consumed by the population and have recently gained attention due to their benefits on health; however, most nutraceuticals are not considered pharmaceutical drugs. Nutraceutical regulation is complex due to the fact that they do not fall under the jurisdiction of pharmaceutical regulatory authorities [21].

Studies have demonstrated therapeutic properties from *M. citrifolia* fruit, such as antioxidant [22], antiulcerogenic [23], anti-inflammatory [24], and anticancer [24,25] properties. Current studies demonstrated noni fruit effects in the prevention or treatment of metabolic diseases in animal models [26,27,28,29,30,31,32]; and, more recently, an important study reported the role of noni fruit juice on type 2 diabetes in humans [33].

Thus, to assess whether *M. citrifolia* fruit displays nutraceutical properties to improve metabolic disturbances present in MetS, the aim of the present study was to evaluate the effects of a crude aqueous extract from *M. citrifolia* Linn. (noni) fruit in the biochemical and histological parameters and in the expression of genes involved in the lipid and glucose metabolism in white adipose tissue and liver in high-fat/high-fructose-fed Swiss mice.

## 2. Materials and Methods

### 2.1. Plant Material

Fruits of *M. citrifolia* were individually selected on the tree and harvested in the municipality of Campo Grande, Mato Grosso do Sul, Brazil (20°27′04.5′′ S and 54°36′04.3′′ W) in March 2017. The tree was properly identified by Flávio Macedo Alves, of the Federal University of Mato Grosso do Sul. A voucher specimen (number 75818) was deposited at the Herbarium Campo Grande/MS (CGMS) of the Federal University of Mato Grosso do Sul, Brazil. This study was approved for the acquisition of samples for genetic access of components and for the associated traditional knowledge (SISGEN—A26D547).

The fruit was harvested while still unripe (pale yellow/stage 3), washed with water to remove superficial dirt, and ripened at room temperature for a day or more, until they reached the normal size and became cream-colored (translucent-grayish/stage 5), indicating physiological maturity and used by the population in folk medicine according to Chan-Blanco et al. (2006) [9].

### 2.2. Preparation of M. citrifolia Fruit Aqueous Extract (AE)

*M. citrifolia* fruit aqueous extract (AE) was prepared in the Laboratory of Natural Products at the Institute of Chemistry in the Federal University of Mato Grosso do Sul, Brazil. Approximately 400 g of the fruit with pulp, seed, and peel [34] was chopped, ground with a blender (Britânia, Goias, Brazil) in 400 mL of deionized water, and filtered with a sieve. The procedure of grinding the content that remained in the sieve with 400 mL water and filtration was repeated three times, to extract the maximum number of present compounds. After that, the AE was lyophilized, fractionated in 1.5 mL cryogenic tubes, and maintained at −80 °C, until the treatments, to avoid oxidation.

### 2.3. Ultra-Fast Liquid Chromatography–Diode Array Detector–Tandem Mass Spectrometry (UFLC–DAD–MS) Analyses

The aqueous extract of *M. citrifolia* (AE) was analyzed through a UFLC LC-20AD coupled to a diode array detector (Shimadzu, Kyoto, Japan) and ESIqTOF microTOF-Q III (Bruker Daltonics, Billerica, MA, USA) mass spectrometer. The ultraviolet (UV) wavelength was monitored between 240 and 800 nm and the mass spectrometer operated in negative ion mode (*m/z* 120–1200). Nitrogen was applied as a nebulizer gas (4 Bar) and dry gas (9 L/min). The capillary voltage was 2.5 kV. The aqueous extract was dissolved in methanol–water (85:15 *v/v*), at a concentration of 1 mg/mL, and eluted through a C-18 solid phase cartridge, followed by filtration on a polytetrafluoroethylene (PTFE) membrane (0.22 µm × 3.0 mm, Millipore). Subsequently, 2 µL was injected into a Kinetex C-18 column (2.6 μm, 150 × 2.2 mm, Phenomenex) protected by a pre-column packed with the same material. The mobile phase was ultrapure water (solvent A) and acetonitrile (solvent B), both containing 0.1% of formic acid (*v/v*), and the gradient elution profile was as follows: 0–2 min, 3% of B; 2–25 min, 3–25% of B; and 25–40 min, 25–80% of B at a flow rate of 0.3 mL/min.

### 2.4. Ethical Statement

All animal experiments were submitted and approved by the Ethics Committee on Animal Use, Federal University of Mato Grosso do Sul (Protocol n° 860/2017).

### 2.5. Acute Oral Toxicity

The acute oral toxicity test of AE was performed in female Swiss mice (*Mus musculus*) [15] based on the OECD (Organization for Economic Co-Operation and Development) Guidelines 425 [35] (OECD, 2008). The animals were divided into two groups (*n* = 10): a control group that received drinking water gavage (*n* = 5) and the treatment group that received AE orally by gavage at a dose of 2000 g/kg (*n* = 5). After treatment, the animals were observed at 30 min, 1 h, 2 h, 3 h, 4 h, 6 h, 12 h, 24 h, and then daily for 14 days. A Hippocratic screening test was conducted to qualify the effects of abnormal morphological and behavioral signs of toxicity. Changes in the body weight, water consumption, and food intake were evaluated on the 1st, 5th, 10th, and 14th days, adapted according to Malone, 1968 [36].

At the end of 14 days, the animals were euthanized (ketamine and xylazine), and the organs (heart lungs, kidneys, liver, spleen, and pancreas) were collected and weighed [37].

### 2.6. Animals and Experiment Design

Swiss adult male mice (*n* = 42, 12 weeks of age) were initially divided into two homogenous groups according to weight: The control group (CT) (*n* = 11) was fed a regular chow diet (70% carbohydrate, 20% protein, and 10% fat) (Nuvilab, Colombo, PR, Brazil), with a caloric content of 3.8 kcal/g [38]; and the high-fat/high-fructose group (HFF) (*n* = 31) was fed a high-fat/high-fructose diet (21.20% carbohydrate; 18.86% protein; 4% soybean oil, 31% lard, and 20% fructose), with a caloric content of 5.45 kcal/g [39] for 9 weeks. After this, the animals of the HFF groups were divided into three homogeneous groups according to weight and were concomitantly supplemented (oral gavage) with *M. citrifolia* aqueous fruit extract (AE) at two different doses: HFFW (HFF + drinking water) (*n* = 10), HFF + AE 250 (HFF + *M. citrifolia* aqueous fruit extract at 250 mg/kg of body weight) (*n* = 11), and HFF + AE 500 (HFF + *M. citrifolia* aqueous fruit extract at 500 mg/kg of body weight) (*n* = 10) (Figure 1). The CT group also received drinking water (CTW) at this stage of the study. The drinking water was used as a vehicle to dissolve the lyophilized AE during the supplementation, and the doses were chosen according to Jambocus et al. (2016) [40]. All groups had ad libitum access to water and food during the experimental period and were kept in an alternating 12 h light/dark cycle, in a temperature-controlled room (22 ± 2 °C).

After 16 weeks of the experiment, the animals were anesthetized with isoflurane (Isoforine, Cristália, Itapira, SP, Brazil) and euthanized by cardiac puncture [41]. The blood and organs were collected for subsequent analysis.

### 2.7. Assessment of Body Fat and Liver Weight

After euthanasia, the fat pads (retroperitoneal, epididymal, perirenal, omental, and mesenteric) and liver were dissected and weighed. The weight of each fat pad (mg) was normalized by the body weight (g) of the animal [38]. The adiposity index was calculated as the total sum of visceral white adipose tissue (g) divided by the final body weight of the animal ×100 and expressed as the percentage of adiposity [41,42,43].

### 2.8. Body Weight and Diet Intake

The mice were weighed two times per week, to evaluate weight changes up to the end of the experiment. The food intake was measured weekly and the feed efficiency index (FEI) [41,44], which refers to the amount of food consumed that can promote body weight gain, was calculated by using Equation (1), where FBW is the final body weight in grams, IBW is the initial body weight in grams, and TF is the total amount of food ingested in grams:FEI = FBW − IBW/TF(1)

### 2.9. Biochemical Analysis

Blood was collected from the inferior vena cava, during euthanasia, to evaluate the serum glucose, serum triglyceride, total cholesterol, high-density lipoprotein cholesterol (HDL-C), and non-HDL cholesterol (non-HDL-C), using the enzymatic colorimetric test (Labtest, Lagoa Santa, Minas Gerais, Brazil).

### 2.10. Histopathological Analysis

Samples of the liver, pancreas, and epididymal adipose tissue were fixed with 10% formalin solution. After fixation, the specimens were dehydrated, embedded in paraffin, cut in a microtome to a thickness of 5 mm each, and stained with hematoxylin and eosin [41,43]. An expert pathologist performed a blind histological analysis of the liver and pancreas and classified the samples according to a score system [45,46]. Pancreas analysis followed the architecture of pancreas evaluation, according to changes in the islets of Langerhans [41,43,47,48].

The adipocyte area of the epididymal adipose tissue was photographed by using a Leica DFC 495 digital camera system (Leica Microsystems, Wetzlar, Germany) integrated into a Leica DM 5500B microscope (Leica Microsystems, Wetzlar, Germany), with a magnification of 200x. The images were analyzed by using the Leica Application Suite software, version 4.0 (Leica Microsystems, Wetzlar, Germany), and a mean area of 100 adipocytes per sample was determined [49].

### 2.11. Oral Glucose Tolerance Test (OGTT)

An oral glucose tolerance test (OGTT) was performed three days prior to initiating treatment with the AE and three days prior to euthanasia. The animals were submitted for 8 h of fasting [41]. The fasting glucose was verified via the flow rate (time 0) using a G-tech glucometer (G-Tech Free, Infopia Co., Ltd., Anyang, Gyeonggi-do city, Korea) with the tail blood. After this, the animals received D-glucose (Vetec, Duque de Caxias, RJ, Brazil) at 2 g/kg of body weight by oral gavage, and the blood glucose was monitored at 15, 30, 60, and 120 min after glucose administration via the tail blood. The area under the curve (AUC) was calculated for each animal, and the mean was calculated for each experimental group [41,50].

### 2.12. Enzyme-Linked Immunosorbent Assay (ELISA)

The levels of plasma insulin were determined according to the Millipore ELISA (enzyme linked immunosorbent assay) Insulin EZRMI-13K kit and performed according to the manufacturer’s instructions (Cat. #EZRMI-13K, EMD Millipore Corporation, St. Charles, MO, USA). Briefly, the blood samples were collected by cardiac puncture and centrifuged at 1900 G for 10 min at 4 °C. The serum was stored at −80 °C, until the day of the assay. The calculated values of the analyzed hormone were based on a standard curve and expressed as ng/mL.

### 2.13. Quantitative Real-Time PCR (qRT-PCR)

Messenger ribonucleic acid (mRNA) expressions were determined by using quantitative real-time PCR (qRT-PCR) in epididymal adipose tissue and liver tissue; briefly, we extracted the total RNA, using TRizol Reagent, according to the manufacturer’s instructions (Thermo Fisher Scientific Inc., Waltham, MA, USA). Two micrograms of the total RNA were reverse transcribed, using a High-Capacity cDNA Reverse Transcription Kit (Thermo Fisher Scientific Inc., Waltham, MA, USA). The relative quantification of the mRNA content was conducted by qRT-PCR, with specific primers and Sybr Green Master Mix (Thermo Fisher Scientific Inc., Waltham, MA, USA). The primers’ sequences are shown in Table 1.

We performed the relative quantification of mRNAs, using the 2-∆∆CT method, and we analyzed the results by applying the Pfaffl equation [51]. We calculated the ∆∆Ct = (Ct of the target gene in the control group—Ct of the target gene in the diet group)/(Ct of housekeeping gene in the control group—Ct housekeeping gene in the diet group). The housekeeping used was ribosomal protein L19 (RPL19). Once the PCR products were produced exponentially, as verified with the primer efficiency curve, we transformed the Ct variation ratios into the fold change, using the formula 2-∆∆Ct [51].

### 2.14. Statistical Analysis

The data were expressed as the mean ± the standard error of the mean (SEM). The statistical differences were determined by using Student’s unpaired *t*-test for independent samples or analysis of variance, followed by Tukey’s post-test (for one-way analysis of variance (ANOVA) or Bonferroni’s post-test (for two-way analysis of variance (ANOVA), to compare more than two groups, using GraphPad Prism, version 8.3.0 software (GraphPad Software, Inc., San Diego, CA, USA). Differences were considered statistically significant when *p* ≤ 0.05.

## 3. Results

### 3.1. Chemical Profile of M. citrifolia Linn. (noni) Fruit Aqueous Extract (AE)

The identification of the components in the aqueous extract of the fruits of *M. citrifolia* was based on analyses of ultraviolet (UV), mass spectrometry (MS), and tandem mass spectrometry (MS^2^) spectrometric data, which were compared with those published in the literature. Seventeen compounds were annotated (Figure 2 and Table 2).

Compound 1 showed an (M-H)^−^ ion at *m/z* 341.1091, compatible with the molecular formula C_12_H_22_O_11_, and was putatively identified as di-*O*-hexoside. Compound 2 was tentatively identified as tri-*O*-hexoside, based on the ions at *m/z* 503.1636 (M-H)^−^ and 549.1679 (M + HCOO)^−^, corresponding to the molecular formula C_18_H_31_O_16_. Accordingly, the MS^2^ spectrum of the formate adduct (*m/z* 549.1679) showed fragment ions at *m/z* 179 and *m/z* 161, attributed to a deprotonated monosaccharide and its dehydrated form, respectively. In addition, the fragment ion at *m/z* 221 agrees with a fragment of non-reducing monosaccharide linked to a 2-hydroxyacetaldehyde [52].

Compound 3, which was annotated as the iridoid deacetylasperulosidic acid, showed a deprotonated ion at *m/z* 389.1090. Aligned with this proposal are the MS^2^ fragment ions depicted in the MS^2^ spectrum [53], as well as the previous report on the isolation of this iridoid from the fruits of *M. citrifolia* [54,55].

Compound 10 was annotated as asperulosidic acid, an iridoid glucoside previously identified in *M. citrifolia* fruits [56], based on the molecular formula C_18_H_24_O_12_, as indicated by the deprotonated ion at *m/z* 431.1217. The fragment ions in the MS^2^ spectrum at *m/z* 165 and 147 agreed with this proposal [57].

Compounds 8, 11, 12, 13, 14, 16, and 17 were annotated as being noniosides (Table 2), considering that their respective accurate masses corresponded to molecular formulae compatible with these compounds, which had already been formerly obtained from *M. citrifolia* fruits [24,58]. Noniosides have been described as fatty acid, fatty alcohol, and hemiterpene glucosides [24,58]. No absorbances above 240 nm were observed in their UV spectra, due to the lack of a suitable chromophore. The only exception was compound 17, which showed a UV absorbance at λ 279 nm, which agrees with noniosides previously isolated from *M. citrifolia* bearing a conjugated fatty acid linked to the sugar moiety [59,60].

Compound 15 showed UV absorbances at λ 291 and 347 nm, and an accurate mass at *m/z* 609.1480, corresponding to the molecular formula C_27_H_30_O_16_. These data are in accordance with those of a flavonol glycoside bearing a luteolin quercetin aglycone [61]. Thus, compound 15 was tentatively identified as rutin, which had been previously isolated and/or identified in *M. citrifolia* fruits [62,63,64].

### 3.2. Acute Oral Toxicity of M. citrifolia (noni) Fruit Aqueous Extract (AE)

We performed the acute oral toxicity of the AE to evaluate whether the AE displayed any sign of toxicity. The results demonstrate no signs of systemic toxicity. There were no changes in the body weight (Appendix A), water consumption (Appendix A), food intake (Appendix A), or excretion of the urine and feces. No changes in the Hippocratic test were observed (Appendix A); for instance, no motor, sensory, and neurobiological changes were observed, as no animals died. The weight of the liver, pancreas, kidneys, lungs, spleen, and heart had no differences between the groups (Appendix A).

### 3.3. Effects of M. citrifolia (noni) Fruit Aqueous Extract (AE) on Body Weight and Food Intake

At the beginning of the experiment, the HFF groups did not display differences in body weight when compared to the CT group (Appendix A). After nine weeks, the HFF group demonstrated increased final body weight and body weight gain compared to the CT. The HFF group presented decreased food and calorie intake and presented higher feed efficiency index when compared to the CT group. These data demonstrated that nine weeks of HFF-diet induction was able to promote body weight gain (Appendix A).

During the ninth week of the diet, the HFF groups were divided, and we began the treatments with the two doses of the *M. citrifolia* (noni) fruit aqueous extract (AE). The effects of the high-fat/high-fructose diet and AE treatment on body weight are demonstrated in Figure 3A,B.

From the 10th to 16th week, the HFF, HFF + AE 250, and HFF + AE 500 groups showed lower food intake in comparison to the CTW group (Figure 3C); nonetheless, the treated groups had lower food and calorie intakes in comparison to the HFFW and CTW groups (Figure 3C,D). The feed efficiency index was increased only in the HFF + AE 250 group compared to CTW group (Figure 3E). The treatment with AE was not able to diminish body weight in mice that were fed with the HFF diet.

### 3.4. Effects of M. citrifolia (noni) Aqueous Extract (AE) on the Visceral Adiposity

We observed that the HFF diet was able to increase the adiposity index in all groups and the two doses of the AE were not able to decrease the adiposity index. The HFF + AE 500 group had higher epididymal fat content in comparison to the CTW group. The retroperitoneal, perirenal, and mesenteric fat contents were higher in HFFW, HFF + AE 250, and HFF + AE 500 in comparison to CTW; however, the HFF + AE 250 group showed increased perirenal fat when compared to HFFW. Our findings demonstrate that the AE treatments were not able to decrease the visceral adiposity. No differences were observed in the liver weight among the groups (Table 3).

### 3.5. Effects of M. citrifolia (noni) Aqueous Extract (AE) on Glucose Tolerance and Systemic Insulin Sensitivity

The groups were divided after 9 weeks of diet induction and the OGTT test was performed prior to the beginning of the treatment of AE (Appendix A). At this point, the HFF group displayed higher blood glucose values when compared to the group that received the regular chow diet (CT), demonstrating that the HFF diet was able to induce glucose intolerance in those animals (Appendix A).

The OGTT performed at the end of the experiment (16th week) indicated that the HFFW and HFF + AE 250 groups remained glucose intolerant when compared to CTW; however, the AE 500 treatment was able to diminish the blood glucose levels in comparison to HFFW, demonstrating that the higher dose of AE improved the glucose tolerance (Figure 4A,B).

We observed that HFFW, HFF + AE 250, and HFF + AE 500 displayed higher insulin serum levels in comparison to CTW (Figure 5A). The same scenario was observed when we calculated the homeostatic model assessment for insulin resistance (HOMA-IR) index in which HFF diet and treatment with AE obtained higher values of HOMA-IR when compared to CTW (Figure 5B). The HFF and HFF + AE 250 groups presented increased values of the homeostatic model assessment for β cells (HOMA-β) index compared to CTW, while the HFF + AE 500 group had no differences among the groups (Figure 5C).

### 3.6. Effects of M. citrifolia (noni) Aqueous Extract on Serum Biochemical Parameters

No differences were observed in the fasting blood glucose among the groups (Figure 6A).

HFFW and the treated groups presented increased total cholesterol (Figure 6B) and triglycerides (Figure 6C) in comparison to CTW, while HFFW and HFF + AE 250 presented higher values of HDL-C in comparison to CTW (Figure 6D); no differences were observed in the non-HDL-C levels among the groups (Figure 6E). Our results demonstrate that the HFF diet was able to enhance the total cholesterol, triglycerides, and HDL-C levels; however, the AE was not able to influence the fasting blood glucose and lipid parameters after the treatment.

### 3.7. Histopathological Analysis of the Epididymal Adipose Tissue, Pancreas, and Liver

The histological analysis of the liver demonstrated a higher prevalence of hepatic steatosis > 5% in the HFW, HFF + AE 250, and HFF + AE 500 groups when compared to the CTW group (*p* = 0.0001). No significant differences were found among the groups that received the HFF diet and treatments (Table 4).

The histological analysis of the pancreas did not demonstrate differences in the variables of Langerhans islets (*p* = 0.29), pancreatic acini (*p* = 0.10), or inflammatory cells among the groups (Table 5). The histological analysis of the liver and pancreas is demonstrated in Figure 7.

The histological analysis of the epididymal adipose tissue demonstrated that the HFFW, HFF+ AE 250, and HFF+AE 500 groups displayed a higher adipocyte area in comparison to the CTW group, while the HFF+AE 500 group had an increased adipocyte area when compared to the HFF+AE 250 group (Figure 8E). Our results show that AE was not able to diminish the size of the adipocytes reaching the control values, and HFF+AE500 worsened the damage caused by the HFF diet.

### 3.8. Effects of M. citrifolia Fruit Aqueous Extract (AE) in the Expression of Genes Involved in Adipocyte Differentiation in White Adipose Tissue and the Lipid and Glycemic Metabolism in the Liver

We compared the mRNA content from genes involved in adipocyte differentiation in white adipose tissue, peroxisome proliferator-activated receptor-γ (PPAR-γ), and peroxisome proliferator-activated receptor-α (PPAR-α) (Figure 9A,B). The HFFW, HFF + AE 250, and HFF + AE 500 groups displayed decreased mRNA content of PPAR-γ compared to CTW (Figure 9A); on the other hand, PPAR-α displayed a lower expression in the HFFW and HFF + AE 250 groups in relation to the CTW group (Figure 9B), while HFF + AE 500 was able to upregulate the PPAR-α expression when compared to HFFW and HFF + AE 250 (Figure 9B) in white adipose tissue.

In the liver, the HFFW and HFF + AE 250 groups displayed increased mRNA content of PPAR-γ when compared to the CTW group; while the HFF + AE 500 group showed significant decreased expression of PPAR-γ in relation to HFFW but not significantly different from CTW (Figure 9C). On the contrary, HFFW and HFF + AE 250 presented decreased mRNA content of PPAR-α in relation to CTW, was only slightly decreased in HFF + AE 500, and was not significantly different from the CTW levels (Figure 9D).

The mRNA content of nuclear transcription factors, such as sterol regulatory element binding protein-1c (SREBP-1c) and carbohydrate response element binding protein (ChREBP), the enzymes glucose-6-phosphatase (G6P) and fatty acid synthase (FAS), and the hepatokine, fetuin-A, were evaluated in the liver (Figure 10). The abundance of SREBP-1c mRNA was higher in HFFW compared to the CTW group (Figure 10A), whereas HFF + AE 500 was lower in relation to HFFW and not different from the CTW group (Figure 10A). No significant differences were observed in the ChREBP (Figure 10B) and FAS (Figure 10C) mRNA expressions among the groups. The G6P mRNA expression was lower in HFFW, HFF + AE 250, and HFF + AE 500 in comparison to the CTW group (Figure 10D). The hepatokine fetuin-A mRNA expression was reduced in the HFF + AE 500 group when compared to all groups (Figure 10E). Our results demonstrate that the AE 500 treatment influenced the regulation of the expression of the nuclear transcription factor, SREBP-1c, and the hepatokine, fetuin-A, in the liver.

## 4. Discussion

The consumption of noni fruit has been discussed regarding its possible health benefits. Products derived from noni fruit are authorized to be commercialized in the form of juice and extracts as encapsulated powder by governmental authorities in European and Asian countries, as well as in the USA [10,11,12]. Even though the fruit is popularly consumed in folk medicine, noni product commercialization is not yet allowed in Brazil [13].

The majority of studies have focused on fermented noni juice or on adding cranberry, blueberry, and grape juice to mask the organoleptic features at the moment of consumption [9,12,18,19,20]. This processing may improve the quality and display a better acceptance from the final consumer [12,19,20]. However, one of the criticisms of consuming products derived from noni fruit is whether the effects of bioactive compounds should be attributed by bioactive compounds per se acting in a synergistic form, or if fermentation or adding other ingredients may alter the quality of the final product. Thus, the use of a crude aqueous noni fruit extract, without fermentation processing or the addition of any type of product to mask noni’s bitter taste, was chosen in our study.

Studies demonstrated that noni fruit juice promoted metabolic effects, such as hypoglycemic and hepatoprotective effects [28,29,30], the reduction in body weight gain [27,28], and antidislipidemic actions [27,29], in diet-induced metabolic dysfunction animal models.

To our knowledge, this is the first study to evaluate the effects of crude aqueous noni fruit extract in high-fat/high-fructose (HFF) fed mice, an animal model that mimics metabolic syndrome in humans [2,65,66].

The results demonstrate that no neurotoxic, behavioral, or mortality effects were produced by AE in the acute toxicity test after or during the post-treatment period (see Appendix A). The HFF diet was able to promote the metabolic abnormalities observed in metabolic syndrome, such as body weight gain and increased visceral adiposity, hyperglycemia, systemic insulin resistance, hypercholesterolemia, hypertriglyceridemia, and hepatic steatosis. The two doses of crude aqueous noni fruit extract (AE) used in the study, AE 250 and AE 500, under the influence of the HFF diet were not able to diminish body weight or the visceral adiposity gain, total cholesterol, triglycerides, insulin levels, systemic insulin resistance, or hepatic steatosis. However, AE 500 was able to inhibit the impaired glucose tolerance promoted by the HFF diet.

The glucose tolerance improvement was also observed by others when fermented noni fruit juice was used to treat diet-induced type 2 diabetes in mice [28], genetic obese type 2 diabetes in mice [30], and type 1 diabetes in rats [67].

Our data demonstrated that AE presents iridoids, noniosides, and flavonoid (rutin); these compounds were previously observed in other studies [24,54,61,68]. Iridoids are found in combination with sugar in most plant species and are named as iridoid glycosides [69]. Iridoids derived from the *M. citrifolia* root extract were previously demonstrated to have hypoglycemic effects in streptozotocin-induced diabetic mice, a typical animal model of type 1 diabetes [54].

A total of 18 novel trisaccharide fatty acid esters (noniosides A-O and others) were isolated specifically from the fruits of *M. citrifolia* [69]; however, there were no studies in diabetes animal models that demonstrated hypoglycemic effects promoted by noniosides from noni fruit. On the other hand, rutin, an important flavonoid found in *M. citrifolia* fruit [68], demonstrated an influence on hypoglycemic parameters [70,71] and also improved dyslipidemia in streptozotocin-induced diabetic rats [70].

There is no evidence demonstrating the indirect influence of bioactive compounds derived from *M. citrifolia* fruit in genes related to lipogenesis. In the human body, the lipogenic pathway is active in at least two important lipogenic tissues, the liver and adipose tissue; however, lipogenesis was demonstrated to be more efficient in the liver compared with in adipose tissue [72]. Deregulations in the lipogenic pathway are associated with diverse pathological conditions. De novo lipogenesis was previously shown to be highly responsive to changes in the dietary regimen, as dietary carbohydrates, particularly fructose, are almost exclusively metabolized by the liver, and the excessive consumption of fructose has been shown to stimulate the deregulation of de novo lipogenesis leading to hepatic lipid accumulation. Hepatic de novo lipogenesis is a metabolic pathway that affects lipid and glucose regulation and plays a role in the development of diabetes, cardiovascular disease, and hepatic steatosis [73].

We evaluated the effects of an HFF diet on the genes involved in the lipid and glucose metabolism and the effect of AE treatment, and we sought for synergistic effects in both white adipose tissue and the liver. Our findings demonstrate that the animals responded differently under the influence of the HFF diet and two doses of AE treatment.

The HFF diet was able to decrease the PPAR-γ and PPAR-α mRNA expression in white adipose tissue. PPAR-γ is majorly expressed in white adipose tissue, is considered a regulator of lipogenesis, and plays a key role in glucose homeostasis and the adipocyte differentiation of fat cells [74]. PPAR-α is predominantly expressed in the liver and white adipose tissue, but not in skeletal muscle, and is a gene involved in fatty acid oxidation that controls fatty acid transport and β-oxidation [75,76] and improves glucose metabolism defects, such as glucose intolerance, hyperglycemia, and insulin resistance [77].

Our results demonstrate that the AE 500 treatment was able to upregulate the PPAR-α mRNA expression but did not change the PPAR-γ mRNA isoform in white adipose tissue. In the liver, AE 500 treatment had a tendency to upregulate the PPAR-α mRNA and downregulate the PPAR-γ mRNA expression. As PPAR-α is able to increase the fatty acid transport and β-oxidation, these could be some of the responsible factors for the glucose tolerance improvement along with the increased lipid accumulation observed in the epididymal fat and liver observed in the AE 500-treated groups.

Shih et al. (2009) also demonstrated that a high-fructose diet promoted the downregulation of PPAR-γ mRNA in white adipose fat deposits and that *Momordica charantia* (bitter melon) extract treatment was able to upregulate the PPAR-γ mRNA expression. The bioactive components from bitter melon responsible for the promotion of hypoglycemic properties included glycosides, saponins, alkaloid, fixed oils, triterpenes, proteins, and steroids [78,79]. In another study, the PPAR-γ mRNA expression did not differ among the groups in white adipose tissue in mice that received a high-fat diet. Further elucidations regarding whether the polyphenols found in the study could activate PPAR-α in white adipose tissue were proposed [80].

Lipogenesis in adipose tissue is less responsive than hepatic de novo lipogenesis to acute or prolonged carbohydrate overfeeding, particularly with fructose. Lipogenesis is an insulin- and glucose-dependent process that is under the control of specific transcription factors, including sterol regulatory binding protein-1c (SREBP-1c), which is activated by insulin, and carbohydrate-responsive element-binding protein (ChREBP), which is activated by glucose in the liver. Insulin induces the maturation of SREBP-1c by a proteolytic mechanism initiated in the endoplasmic reticulum, and consequently, SREBP-1c activates glycolytic gene expression, promoting the glucose metabolism, as well as lipogenic genes in conjunction with ChREBP [81,82].

SREBP-1c and ChREBP respond differently under the influence of fructose or glucose supplementation in a high-fat diet. While SREBP-1c is upregulated after fructose stimulation, leading to hepatic lipogenesis and insulin resistance, ChREBP is upregulated and activated by glucose supplementation [83]. As observed in our data, an HFF diet upregulated the SREBP-1c mRNA expression and the AE 500 treatment downregulated its expression; however, no differences were observed in the ChREBP mRNA expression, indicating that the effects on inhibiting impaired glucose tolerance promoted by AE 500 treatment may be associated with the SREBP-1c pathway and not the ChREBP pathway. In the liver, research reported that ChREBP is under the influence of G6P [84,85]; hence, the lack of ChREBP activation in our experimental groups suggest that this was due to the low G6P expression.

On the contrary, SREBP-1c is primarily involved in the regulation of genes related to fatty acid synthesis, such as fatty acid synthase (FAS), a marker for lipogenesis and energy metabolism. However, the high-carbohydrate/high-fat diet did not display differences in the FAS mRNA expression in mice liver [86]. Huang and Lin (2012) observed that a high-fructose diet promoted a higher FAS gene expression and green, black, and pu-erh tea leaves, which contain catechins, caffeine, and theanine, led to a suppression of the FAS gene in the liver. We did not observe any differences in the hepatic FAS gene expression among the groups, suggesting no influence from the HFF diet or AE treatments [87].

The hepatic fetuin-A mRNA expression was shown to correlate with the hepatic mRNA levels of key enzymes in the lipid and glucose metabolism [88]. This hepatokine is a liver-derived protein and, together with free fatty acids, induces apoptotic signals in the beta islet cells of the pancreas, reducing the secretion of insulin and further exacerbating diabetes type 2 [89]. Jung et al., (2013) observed that the incubation of hepatocytes with palmitate, a model of NAFLD in vitro, induced upregulation of the fetuin-A expression in hepatocytes, accompanied by triglyceride accumulation and the induction of SREBP-1c. The knock-down of fetuin-A by silencing RNA (siRNA) restored these changes [88]. Another study observed that fetuin-A induced mammalian target of rapamycin (mTOR) phosphorylation, which has been reported to induce the activation of SREBP-1c expression, thus, demonstrating the correlation of activating fetuin-A indirectly activating SREBP-1c [90].

Our findings demonstrate that the AE 500 treatment was able to reduce fetuin-A mRNA expression in relation to all groups, along with the downregulation of SREBP-1c mRNA, under the influence of the HFF diet. Although we observed that the AE treatments had no influence on most of the metabolic parameters, in our diet-induced metabolic syndrome animal model, the AE 500 treatment was shown to improve the glucose tolerance. This effect could be associated with the indirect influence of bioactive compounds found in noni extract, such as iridoids, noniosides, and rutin, acting in a synergistic manner in PPAR-α, PPAR-γ, SREBP-1c, and fetuin-A gene expression, which are involved in the lipid and glycemic metabolism in the liver and in white adipose tissue.

Certain considerations are important to be highlighted in our study, and one is that the introduction of an HFF diet at weaning could yield a more pronounced metabolic syndrome phenotype compared with the introduction of an HFF diet in young adults (12 weeks of age). The evaluation of an extended period of treatment or the use AE as a supplementation at the beginning of diet induction would be of value. Further studies are necessary to evaluate the role of the AE 500 treatment’s influence in the SREBP-1c pathway, for instance, related to mammalian target of rapamycin (mTOR) phosphorylation and serine/threonine kinase (S6k), as well as to evaluate the influence of AE 500 on PPAR expression in white adipose tissue

## 5. Conclusions

Neither dose of AE exhibited neurotoxic, behavioral, nor mortality effects in the acute toxicity test, after or during the post-treatment period. Iridoids, noniosides, and the flavonoid rutin were tentatively identified in AE through analyses of the UFLC–DAD–MS data. Neither dose of AE influenced the majority of the metabolic indicators studied (including weight loss, the total cholesterol, triglycerides, non-HDL, HDL-C, and the insulin serum levels) or the histological parameters in the liver, pancreas, and epidydimal adipose tissue. However, the higher dose of AE (AE 500) was shown to be effective in improving the glucose tolerance in the OGTT test, which could be associated with the influence of the PPAR-α expression in adipose tissue and the PPAR-γ, PPAR-α, SREBP-1c, and fetuin-A expression in the liver. Further studies can explore how noni extract influences PPARs in adipose tissue and the SREPB-1c pathway in the liver.

## Figures and Tables

**Figure 1 nutrients-12-03439-f001:**
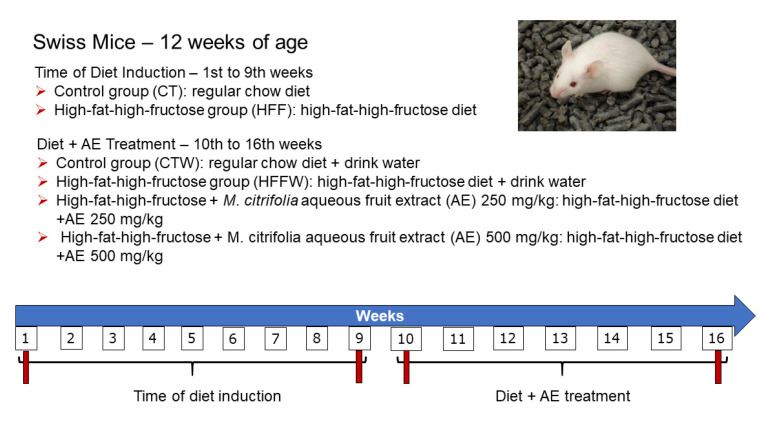
Schematic figure of the experimental design to assess the time of diet induction (from the 1st to 9th weeks) and diet + *Morinda citrifolia* aqueous fruit extract treatment (AE) (the 10th to 16th weeks).

**Figure 2 nutrients-12-03439-f002:**
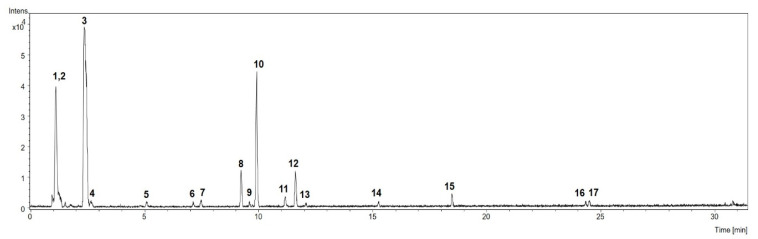
The base peak chromatogram (BPC) obtained by UFLC–DAD–MS of *M. citrifolia* aqueous fruit extract (AE), in negative mode. The identification of chromatographic peaks is described in Table 2.

**Figure 3 nutrients-12-03439-f003:**
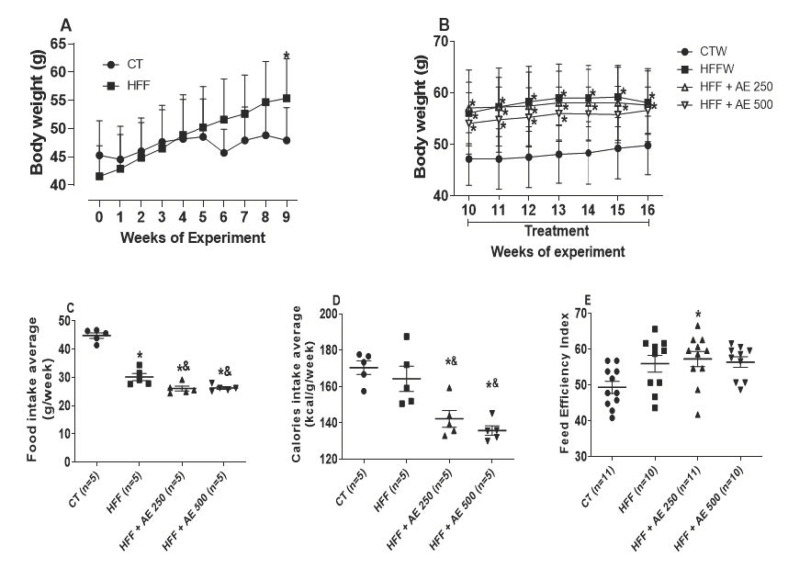
Effects of the high-fat/high-fructose diet and AE on body weight. (**A**) The body weight of animals fed on the regular chow diet (CT, *n* = 11) and on the high-fat/high-fructose diet (HFF, *n* = 31) for nine consecutive weeks (0: initial weight); (**B**) the body weight of animals of the full black circle: CTW diet (regular chow diet + drinking water, *n* = 11); full black quadrilateral: HFFW diet (high-fat/high-fructose diet + drinking water, *n* = 10); black up-pointing triangle: HFF + AE 250 diet (HFF + *M. citrifolia* fruit aqueous extract of 250 mg/kg of body weight, *n* = 11); and black down-pointing triangle: HFF + AE 500 diet (HFF + *M. citrifolia* fruit aqueous extract of 500 mg/kg of body weight, *n* = 10) for seven weeks (10th to 16th week); (**C**) the food intake average for 16 weeks; (**D**) the calorie intake average for 16 weeks; (**E**) the feed efficiency index for 16 weeks. The results are expressed as the mean ± standard error of the mean (SEM). Two-way ANOVA, followed by the Bonferroni post-test. * *p* ≤ 0.05 vs. CTW. & = *p* ≤ 0.05 vs. HFFW.

**Figure 4 nutrients-12-03439-f004:**
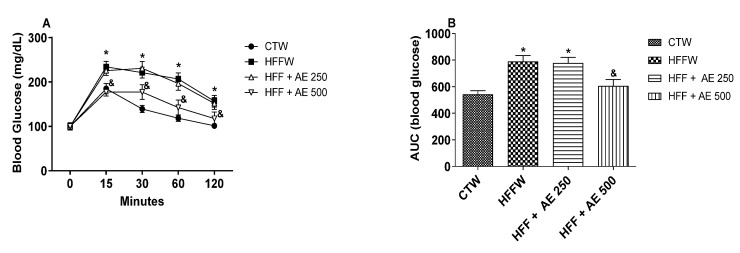
Evaluation of the glycemic profile at the end of the treatment with AE. (**A**) The oral glucose tolerance test at the end of the treatment (16th week); (**B**) the area under the curve (AUC) of the blood glucose of animals evaluated at the end of the treatment (16th week) of CTW (CT + drinking water, *n* = 11); HFFW (high-fat/high-fructose diet + drinking water, *n* = 11); HFF + AE 250 (HFF + *M. citrifolia* fruit aqueous extract of 250 mg/kg of body weight, *n* = 11); and HFF + AE 500 (HFF + *M. citrifolia* fruit aqueous extract of 500 mg/kg, *n* = 10) groups. The results are expressed as the mean ± SEM. * = *p* ≤ 0.05 vs. CTW. & = *p* ≤ 0.05 vs. HFFW. Two-way ANOVA, followed by the Bonferroni post-test for the oral glucose tolerance test. ANOVA, followed by the Tukey post-test for the AUC.

**Figure 5 nutrients-12-03439-f005:**
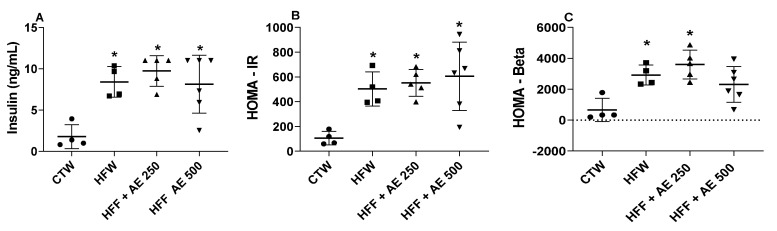
Evaluation of the insulin levels, homeostatic model assessment for insulin resistance (HOMA-IR), and homeostatic model assessment for β cells (HOMA-β). (**A**) The insulin serum levels (ng/mL). (**B**) HOMA-IR; (**C**) HOMA-β of animals evaluated at the end of the treatment (16th week) of full black circle: CTW (CT + drinking water, *n* = 4), full black quadrilateral: HFFW (high-fat/high-fructose diet + drinking water, *n* = 4); black up-pointing triangle: HFF + AE 250 (HFF + *M. citrifolia* fruit aqueous extract of 250 mg/kg of body weight, *n* = 5); and black down-pointing triangle: HFF + AE 500 (HFF + *M. citrifolia* fruit aqueous extract of 500 mg/kg, *n* = 6) groups. The results are expressed as the mean ± SEM. * = *p* ≤ 0.05 vs. CTW. ANOVA, followed by the Tukey post-test.

**Figure 6 nutrients-12-03439-f006:**
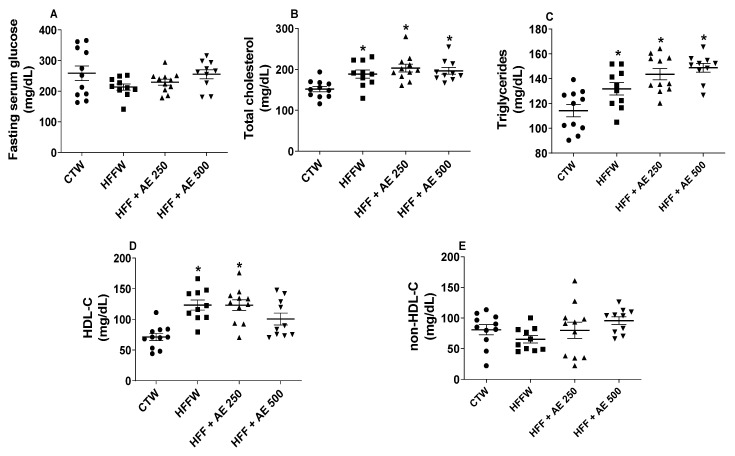
The effects of the high-fat/high-fructose diet and *M. citrifolia* (noni) aqueous extract (AE) on the serum biochemical parameters. (**A**) The fasting blood glucose (mg/dL); (**B**) total cholesterol (mg/dL); (**C**) triglycerides (mg/dL); (**D**) high-density lipoprotein cholesterol (HDL-C) (mg/dL); (**E**) non-HDL-C (mg/dL) of full black circle: CTW (CT + drinking water, *n* = 11), full black quadrilateral: HFFW (high-fat/high-fructose diet + drinking water, *n* = 10); black up-pointing triangle: HFF + AE 250 (HFF + *M. citrifolia* fruit aqueous extract of 250 mg/kg of body weight, *n* = 11); and black down-pointing triangle: HFF + AE 500 (HFF + *M. citrifolia* fruit aqueous extract of 500 mg/kg groups, *n* = 10) for seven weeks (10th to 16th weeks). The results are expressed as the mean ± SEM. * = *p* ≤ 0.05 vs. CTW. ANOVA, followed by the Tukey post-test.

**Figure 7 nutrients-12-03439-f007:**
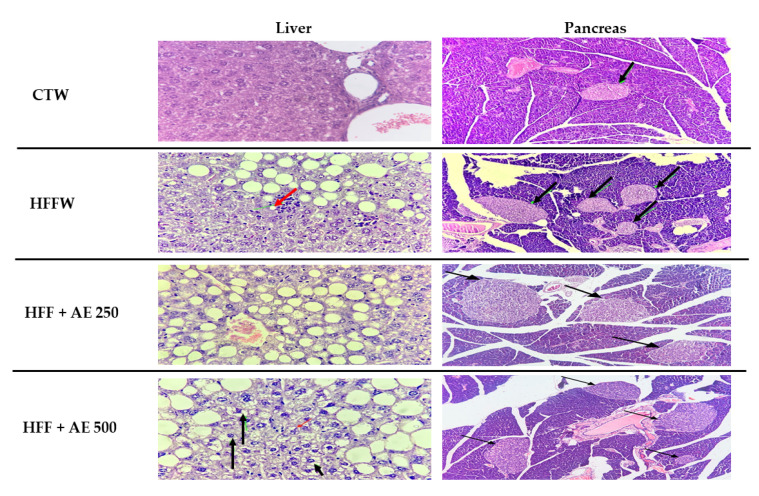
Histological analysis of the liver (red arrow indicates lobular inflammation, and black arrows indicate ballooning), 400× magnification, bar scale: 100 µm and pancreas (black arrows indicate Langerhans islets), 100× magnification, bar scale: 100 µm of CTW (CT + drinking water), HFFW (high-fat/high-fructose diet + drinking water), HFF + AE 250 (HFF + *M. citrifolia* fruit aqueous extract of 250 mg/kg of body weight), HFF + AE 500 (HFF + *M. citrifolia* fruit aqueous extract of 500 mg/kg).

**Figure 8 nutrients-12-03439-f008:**
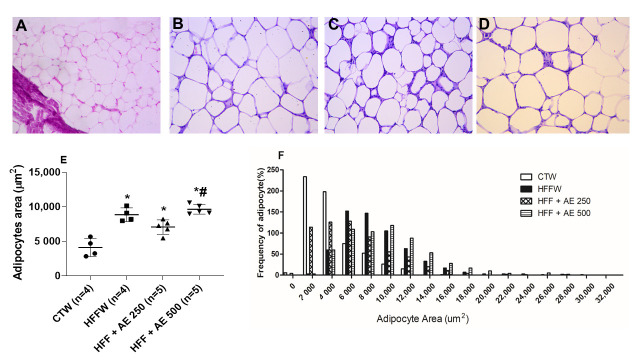
Histological analysis of the epididymal adipose tissue of each group. (**A**) CTW (CT + drinking water); (**B**) HFFW (high-fat/high-fructose diet + drinking water); (**C**) HFF + AE 250 (HFF + *M. citrifolia* fruit aqueous extract of 250 mg/kg of body weight); (**D**) HFF + AE 500 (HFF + *M. citrifolia* fruit aqueous extract of 500 mg/kg) 200 x magnification. Bar scale: 100 µm; (**E**) adipocyte area (µm^2^) of full black circle: CTW (CT + drinking water); full black quadrilateral: HFFW (high-fat/high-fructose diet + drinking water); black up-pointing triangle: HFF + AE 250 (HFF + *M. citrifolia* fruit aqueous extract of 250 mg/kg of body weight); black down-pointing triangle: HFF + AE 500 (HFF + *M. citrifolia* fruit aqueous extract of 500 mg/kg); (**F**) graph of the frequency of distribution of adipocytes (%). The results are expressed as the mean ± SEM. * = *p* ≤ 0.05 vs. CTW; ^#^ = *p* ≤ 0.05 vs. HFF + AE 250. ANOVA followed by the Tukey post-test.

**Figure 9 nutrients-12-03439-f009:**
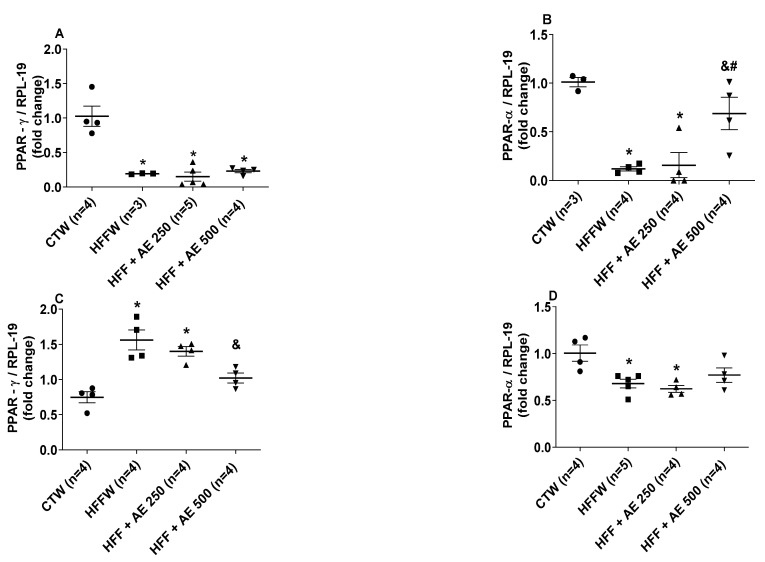
Effects of the HFF diet and *M. citrifolia* (noni) aqueous extract (AE) in the mRNA content of metabolic genes in the epididymal fat and liver. (**A**) Peroxisome proliferator-activated receptor-γ (PPAR-γ) in epididymal adipose tissue; (**B**) peroxisome proliferator-activated receptor-α (PPAR-α) in epididymal adipose tissue; (**C**) peroxisome proliferator-activated receptor-γ (PPAR-γ) in the liver; (**D**) peroxisome proliferator-activated receptor-α (PPAR-α) in the liver of full black circle: CTW (CT + drinking water), full black quadrilateral: HFFW (high-fat/high-fructose diet + drinking water); black up-pointing triangle: HFF + AE 250 (HFF + *M. citrifolia* fruit aqueous extract of 250 mg/kg); and black down-pointing triangle: HFF + AE 500 (HFF + *M. citrifolia* fruit aqueous extract of 500 mg/kg) groups. The results are expressed as the mean ± SEM. * = *p* ≤ 0.05 vs. CTW; ^&^ = *p* ≤ 0.05 vs. HFFW; ^#^ = *p* ≤ 0.05 vs. HFF+AE 250. ANOVA followed by the Tukey post-test.

**Figure 10 nutrients-12-03439-f010:**
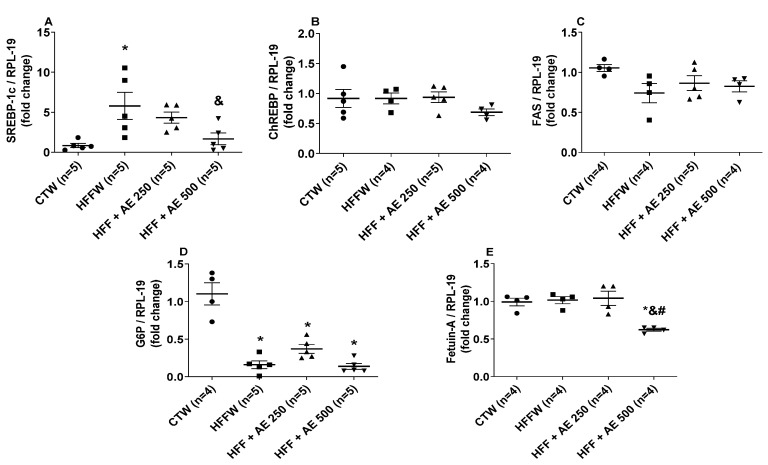
The effects of a high-fat/high-fructose diet and *M. citrifolia* (noni) aqueous extract (AE) in the expression of metabolic genes in the liver (**A**) sterol regulatory element binding protein-1c (SREBP-1c); (**B**) carbohydrate response element binding protein (ChREBP); (**C**) fatty acid synthase; (**D**) glucose-6-phosphatase (G6P); (**E**) fetuin-A of full black circle: CTW (CT + drinking water), full black quadrilateral: HFFW (high-fat/high-fructose diet + drinking water); black up-pointing triangle: HFF + AE 250 (HFF + *M. citrifolia* fruit aqueous extract of 250 mg/kg); and black down-pointing triangle: HFF + AE 500 (HFF + *M. citrifolia* fruit aqueous extract of 500 mg/kg) groups. The results are expressed as the mean ± SEM. * = *p* ≤ 0.05 vs. CTW; ^&^ = *p* ≤ 0.05 vs. HFFW; ^#^ = *p* ≤ 0.05 vs. HFF + AE 250. ANOVA, followed by the Tukey post-test.

**Table 1 nutrients-12-03439-t001:** Sequence of the oligonucleotides used in qRT-PCR.

Gene	Sequence	Amplicon (bp)	Accession Number
Peroxisome Proliferator Activated Receptor Gamma (PPAR-γ)	Fwd: ATCTTAACTGCCGGATCCAC Rev: CAAACCTGATGGCATTGTGAG	102	NM_001127330.2
Peroxisome Proliferator Activated Receptor Alpha (PPAR-α)	Fwd: TGCAATTCGCTTTGGAAGAA Rev: CTTGCCCAGAGATTTGAGGT	118	NM_011144.6
Fat Acid Synthase (FAS)	Fwd: GATTCGGTGTATCCTGCTGTC Rev: CATGCTTTAGCACCTGCTGT	95	NM_007988.3
Glucose-6-Phosphatase (G6P)	Fwd: CCGGATCTACCTTGCTGCTC Rev: GCATTGTAGATGCCCCGGAT	105	NM_008061.4
Fetuin-A	Fwd: GGAGATTTCCCGGGCTCAAA Rev: TGCAGTACAGTCAGTGGCAG	82	NM_001276450.1
Carbohydrate Response Element Binding Protein (ChREBP)	Fwd: CAGCATCGATCCGACACTCA Rev: GGCCTTTGAAGTTCTTCCACT	96	NM_021455.5
Sterol Regulatory Element Binding Transcription Factor 1-c (SREBP1-c)	Fwd: TGACGGAGACAGGGAGTTCT Rev: CAGAGAAACTGCAAGCAGGA	95	NM_001313979.1
Ribosomal Protein L19 (RPL19)	Fwd: CAATGCCAACTCCCGTCA Rev: GTGTTTTTCCGGCAACGAG	102	NM_009078.2

Sequence of the oligonucleotides: Fwd (Forward); Rev (Reverse).

**Table 2 nutrients-12-03439-t002:** Compounds tentatively identified in *M. citrifolia* aqueous fruit extract (AE).

N°	Rt (min)	Compound	Molecular Formula *	MS (^−^)	MS ^2^	UV λ_max_ (nm)
1	1.2	di-O-hexoside	C_12_H_22_O_11_	341.1091 (M − H)^−^	-	-
2	1.2	tri-O-hexoside	C_18_H_32_O_16_	503,1636 (M − H)^−^ 549.1679 (M + HCOO)^−^	549.1679: 221, 179, 161	-
3	2.4	Deacetylasperulosidic acid	C_16_H_22_O_11_	389.1090 (M − H)^−^	389.1090: 209, 165	-
4	2.7	Unknown	C_17_H_26_O_12_	421.1323 (M − H)^−^	-	-
5	5.1	Unknown	C_16_H_28_O_12_	411.1500 (M − H)^−^	-	-
6	7.1	Unknown	C_15_H_26_O_11_	381.1405 (M − H)^−^	-	-
7	7.5	Unknown	C_15_H_26_O_11_	381.1407 (M − H)^−^	-	-
8	9.2	Nonioside (hemiterpene disaccharide)	C_17_H_30_O_11_	409.1735 (M − H)^−^	-	-
9	9.6	Unknown	C_16_H_24_O_10_	375.1285 (M − H)^−^	-	-
10	9.9	Asperulosidic acid	C_18_H_24_O_12_	431.1217 (M − H)^−^	431.1217: 165, 147	-
11	11.2	Nonioside	C_16_H_28_O_11_	395.1593 (M − H)^−^, 441.1623 (M + HCOO)^−^	-	-
12	11.6	Nonioside (hemiterpene disaccharide)	C_16_H_28_O_10_	379.1636 (M − H)^−,^ 425.1692 (M + HCOO)	-	-
13	12.1	Nonioside (hemiterpene disaccharide)	C_16_H_28_O_10_	379.1647 (M − H)^−,^ 425.1679 (M + HCOO)	-	-
14	15.3	Nonioside (fatty acid ester disaccharide)	C_18_H_32_O_12_	439.1831 (M − H)^−,^	-	-
15	18.5	Rutin	C_27_H_30_O_16_	609.1480 (M − H)^−^	-	291, 347
16	24.3	Nonioside (fatty acid ester disaccharide)	C_20_H_36_O_12_	467.2161 (M − H)^−^	-	-
17	24.5	Nonioside (fatty acid ester disaccharide)	C_22_H_34_O_12_	489.2001 (M − H)^−^	-	279

Rt: retention time; UV: ultraviolet; MS: mass spectrometry; MS^2^: tandem mass spectrometry. * All molecular formulas were determined from the accurate mass, considering a mass error lower than 8 ppm.

**Table 3 nutrients-12-03439-t003:** The effects of AE on the fat pads, adiposity index, and liver weight.

Parameter			Groups	
	CTW (*n* = 11)	HFFW (*n* = 10)	HFF + AE 250 (*n* = 11)	HFF + AE 500 (*n* = 10)
Epididymal fat (mg/g)	28.68 ± 2.69	42.10 ± 3.77	39.71 ± 3.95	48.67 ± 3.66 *
Retroperitoneal fat (mg/g)	8.13 ± 0.45	16.81 ± 1.96 *	16.09 ± 1.28 *	16.56 ± 1.00 *
Perirenal fat (mg/g)	6.79 ± 0.71	10.75 ± 1.14 *	15.90 ± 0.94 *^&^	14.27 ± 1.17 *
Omental fat (mg/g)	0.42 ± 0.07	1.04 ± 0.25	0.71 ± 0.17	0.62 ± 0.21
Mesenteric fat (mg/g)	18.47 ± 1.04	29.20 ± 2.86 *	34.14 ± 1.55 *	34.29 ± 1.3 4 *
Liver (mg/g)	38.11 ± 0.60	42.62 ± 3.87	42.83 ± 2.55	40.33 ± 2.92
Adiposity index (%)	6.25 ± 0.46	9.99 ± 0.62 *	10.65 ± 0.49 *	11.44 ± 0.61 *

CTW: regular chow diet + drinking water. HFFW: high-fat/high-fructose diet + drinking water. HFF + AE 250: HFF + *M. citrifolia* fruit aqueous extract of 250 mg/kg of body weight. HFF + AE 500: HFF + *M. citrifolia* fruit aqueous extract of 500 mg/kg of body weight for seven weeks (10th to 16th week). The results are expressed as the mean ± SEM. ANOVA followed by Tukey post-test. * *p* ≤ 0.05 vs. CTW. & = *p* ≤ 0.05 vs. HFFW.

**Table 4 nutrients-12-03439-t004:** Results for the changes observed in the livers of animals from each experimental group.

Variable			Groups	
Liver Changes	CTW (*n* = 10)	HFFW (*n* = 10)	HFF + AE 250 (*n* = 10)	HFF + AE 500 (*n* = 11)
Steatosis *(*p* < 0.0001)				
Score: 0 = none; 1 = light, 2 = moderate; 3 = severe	0.0 ± 0.0	2.0 ± 0.15 *	1.9 ± 0.18 *	2.273 ± 0.14 *
Steatosis Localization	None	Zone 3	Zone 3	Zone 3
Microvesicular Steatosis (*p* = 0.09)				
0 = absent; 1 = present (number of mice)	0	0	1 (8 out of 10)	0
Lobular Inflammation*(*p* < 0.0001)				
Score: 0 = Absent; 1 = < 2 focus/field; 2 = 2-4 focus/field	0.1 ± 0.1	0.9 ± 0.18 *	1.0 ± 0 *	1.182 ± 0.18 *
Ballooning *(*p* = 0.05)				
Score: 0 = Absent; 1 = Few cells, 2 = Many cells	1.0 ± 0.15	1.7 ± 0.15 *	1.7 ± 0.15 *	1.64 ± 0.15 *
Mallory’s Hyaline *(*p* = 0.07)				
Score: 0 = Absent; 1 = Rare, 2 = Several	0.2 ± 0.13	0.3 ± 0.2	1.1 ± 0.28 *	1.1 ± 0.25 *
Apoptosis (*p* = 0.45)				
Score: 0 = Absent; 1 = Present	0.1 ± 0.1	0.6 ± 0.16	0.5 ± 0.17	0.36 ± 0.15
Glycogenate Nucleus (*p* = 0.12)				
Score: 0 = Absent/Rare; 1 = Some	0.6 ± 0.16	0.1 ± 0.1	0.5 ± 0.16	0.36 ± 0.15

CTW (CT + drinking water), HFFW (high-fat/high-fructose diet + drinking water), HFF + AE 250 (HFF + *M. citrifolia* fruit aqueous extract of 250 mg/kg of body weight), HFF + AE 500 (HFF + *M. citrifolia* fruit aqueous extract of 500 mg/kg). Inferential statistical analysis due to the absence of values in the analyzed categories. The data are expressed as the mean + SEM of the relative scores. Analysis of one-way variance (ANOVA) and Bonferroni post-test. * = *p* ≤ 0.05 vs. CTW.

**Table 5 nutrients-12-03439-t005:** Results for the changes observed in the pancreas of animals from each experimental group.

Variable			Groups	
Pancreas Changes	CTW (*n* = 11)	HFFW (*n* = 10)	HFF + AE 250 (*n* = 11)	HFF + AE 500 (*n* = 10)
Islet of Langerhans (*p* = 0.11)				
Score: 0 = No change; 1 = Discrete Atrophy; 2 =Atrophy	0.3 ± 0.15	1.25 ± 0.3	0.62 ± 0.3	0.7 ± 0.29
Pancreatic acini (*p* = 0.09)	0.0 ± 0.0	0.2 ± 0.1	0.0 ± 0.0	0.0 ± 0.0
Score: 0 = No change; 1 = Necrosis/Atrophy				
Inflammatory cells (*p* = 0.33)	0.0 ± 0.0	0.0 ± 0.0	0.1 ± 0.1	0.0 ± 0.0
Score: 0 = No change; 1 = Perinsulitis				

CTW (CT + drinking water), HFFW (high-fat/high-fructose diet + drinking water), HFF + AE 250 (HFF + *M. citrifolia* fruit aqueous extract of 250 mg/kg of body weight), HFF + AE 500 (HFF *+ M. citrifolia* fruit aqueous extract of 500 mg/kg). Inferential statistical analysis due to the absence of values in the analyzed categories. The data are expressed as the mean + SEM of the relative scores. Analysis of one-way variance (ANOVA) and Bonferroni post-test.

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
