# Peer review of "Therapeutic Effects of Morinda citrifolia Linn. (Noni) Aqueous Fruit Extract on the Glucose and Lipid Metabolism in High-Fat/High-Fructose-Fed Swiss Mice"

_nutrients, 2020, doi:10.3390/nu12113439_

Round 1

Reviewer 1 Report

This study investigated the effect of Morinda citrifolia aqueous extract on glucose and lipid metabolism on high fat-high fructose (HFHF) fed Swiss mice. For the study, a subset of animals was experimentally exposed to HFHF diet for 9 weeks. After this period of time, the experimental group was further divided into three groups, two of which receiving two different doses of the aqueous extract (250 or 500 mg/kg body weight, respectively). All three experimental groups were compared to the control group, and assessed for a variety of biological and histological parameters. The results reported in the study indicate that the higher dose of aqueous extract improved glucose tolerance (as blood glucose level and resulting AUC) as well asPPAR-gamma, SREBP1c, and fetuin-A in the liver of the animals. In contrast, neither dose had an effect on pancreas histology or other metabolic parameters. The conclusion of the authors is that aqueous extract of the Morinda citrifolia may have hypoglycemic effect by affecting 'the expression of genes involved in hepatic de novo lipidogenesis (verbatim)'.

Comments

Overall, the study appears to be properly conducted, and the conclusions of the authors are supported by the data presented in the manuscript. A few points, however, need to be clarified. 

  1. How do the two doses of aqueous extract compare to doses that can be achieved in-vivo by individuals consuming the fruit or the fruit extract?
  2. Why were female animals used for the study? I am just curious about the rationale for using female animals, as this raises the question whether the aqueous extract has any effect on the circulating level of estrogens and progesterone and consequently on the animal metabolism
  3. Did the authors investigated any direct effect on insulin signaling (Akt activation, PIP3, etc.)? All the information about insulin effectiveness is indirect, based on blood glucose level and HOMA, or systemic metabolic parameters and not direct such as phosphorylation of key signaling components. This could be particularly important for the effect observed in liver metabolism, as different tissues (or different enzymes) could respond differently. 

Reviewer 2 Report

In this report, Inada et al test the therapeutic efficacy of two doses of noni aqueous extract on a diet-induced model of metabolic syndrome. They have identified Seventeen compounds including iridoids, noniosides and the flavonoid rutin. In addition, they state that: “The higher dose of AE (AE 500 mg/kg) demonstrated to improve glucose tolerance; however, both doses did not have effects in other metabolic and histological parameters. AE 500 mg/kg downregulated PPAR-γ, SREBP-1c and fetuin-A mRNA and had a tendency to up-regulate PPAR-α mRNA in liver, suggesting that the hypoglycemic effects could be associated by the influence of AE 500 mg/kg in the expression of genes involved in hepatic de novo lipogenesis.” Overall, the manuscript contains interesting observations, however there are several points for improvements, including: study design, statistical tests used, data presentation, discussion of results and some additional experimental questions. Please find below a comprehensive review of your manuscript –which has the sole intention of improving the scientific soundness and impact of the work presented.

1) Methods: from my understanding, diet induction began at 12 weeks of age, extract treatment began at week 21 and ended when mice were 28 weeks of age. An image depicting the study design would greatly improve readers’ understanding of the work. In addition, for future studies, please bear in mind that introducing the HFF diet at weaning might yield a more pronounced metabolic phenotype. It may also be a closer model to humans, as there are increasing reports of unhealthy diets beginning in childhood and adolescence that extend throughout adulthood.

2) Methods: Was the pathologist blinded for the histopathological analyses? This should be clearly stated and, if not (i.e. if the person evaluating the samples knew to which group they belonged) this should be stated as a limitation. In addition, statistical tests employed (chi-square) are unsuitable. I have tried to replicate data in Table 6 and 7 using Graphpad Prism 8, but the programme stated that “Chi-square calculations are only valid when all expected values are greater than 1.0 and at least 20% of the expected values are greater than 5. These conditions have not been met, and thus the chi-square calculations are not valid.” My suggestion is to condensate some analyses – for instance, use a simpler NAFLD score for liver (e.g. https://doi.org/10.1017/S2040174418000284 *). Or present this as semi-quantitative data. Moreover, it seems extract treatment has exacerbated some liver alterations – are ALT/AST/GGT altered?

* Disclaimer: I am not a co-author on this paper, neither am I asking for it to be cited.

3) To improve transparency of data, please provide individual values in graphs – e.g. scatter plot or bar + individual points.

4) Is it possible to quantify the components of the extract to identify which is the most likely active compound?

5) For the toxicity data (Supplementary Figure), the authors claim that “no changes in the Hippocratic test were observed, for instance motor, sensory and neurobiological changes, as no animals died.” However they do not provide this data. Please provide detailed motor, sensory and neurobiological data to sustain your claims, otherwise, please tone down this section.

6) Table 3 is not essential – could be added as supplementary

7) Table 4: Please provide data on body weight, instead of weight gain. If possible, please provide a time-course of body weight. It would be useful to present a figure with the phenotype for the model+extract treatment, including: evolution of body weight and food intake/efficiency as graphs, as these are important datasets for the study.

8) Table 5: It seems AE treatment led to increased fat accumulation of some fat pads (perirenal for AE250, epidydimal for AE500). Why is this? Please provide detailed discussion or address this experimentally (for instance, are adrenergic genes downregulated?).

9) Figure 2: Graphs A and B are not essential – could be added as supplementary

10) It would be clearer for the reader to present on the same graph data on: OGTT, fasting glycaemia, insulinaemia, HOMA, circulating lipids and TyG (insulin resistance marker, please calculate using glycaemia and triglyceredaemia). Are leptin and adiponectin levels affected, bearing in mind increased specific adipose tissue accumulation in AE-treated groups?

11) The authors state that AE treatment had a hypoglycemic effect, which is not correct. The only phenotypic effect observed was improved glucose tolerance, which could be interpreted as an anti-hyperglycaemic effect (slightly different mechanism than hypoglycaemic). Question: is it possible that extract treatment prevents glucose absorption? Perhaps ipGTT could indirectly answer this. Discussion may be needed to address this possibility.

12) Table 6 and 7: as pointed above, please provide either a different method of analyses or statistical test. In addition, please present better representative images. Pancreas images for AE250 and AE500 are not acceptable.

13) Please quantify triglycerides, cholesterol and total fat in liver – this could be a good link with PCR data focused on genes of de novo lipogenesis.

14) Figure 6: A frequency distribution graph of adipocyte area may be useful to understand if HFF or HFF+AE treatment induced different subpopulations of adipocytes.

15) Figures 7 and 8: Please condensate these on a single figure and clearly state data on fatpad and data on liver.

16) Figure 9: why was the analysis focused on liver tissue instead of epididymal fat? PCR data attempted to explain the phenotypic observation of extract treatment, which were: improved glucose tolerance + possibly enhanced adiposity. Measurement of genes related to glucose tolerance or those involved in adrenergic output in fat could greatly improve the manuscript.

17) If AE has rutin in its composition and if rutin is a potent and selective inhibitor of PDIA1: are PDIA1 levels affected by diet induction and/or extract treatment? ER stress may well be another component of the effects observed in this manuscript and could be added in discussion, if authors are unable to provide experimental proof of this.

18) Quantification of protein levels using western blot could also improve the manuscript – could focus on only those in which gene expression was differentially expressed (PPARs, fetuin-a, srebp-1c).

19) If iridoids affect the PI3K pathway – have you looked at this in your samples?

20) The Discussion section is very confusing and does not provide a comprehensive interpretation of the data. Please focus on your most important results and take the reader through them, pointing out what has been shown previously in the literature as well as the implications of your work to the scientific field. As it is now, the discussion is rather clunky and there are a lot of citations of papers that do not contribute to the interpretation of your data – please trim down cited work and focus on discussing key points.

21) A limitations paragraph may be useful to state which datasets could be improved and how.

22) Please provide n number for each dataset presented on respective figure/table legend.

23) The paper needs moderate review of the English language.

I hope that the suggestions above are useful to the authors and are able to improve the manuscript - both its impact and clarity of ideas.

Reviewer 3 Report

A well prepared manuscript by the authors.

I have a few comments on the content:

  1. There are many grammar mistakes throughout the text which needs to be addressed
  2. The quality of the stained images is low and would be better to replace with higher quality and less variation of staining.
  3. There is an extra 's' in word 'levels' (line 431)
  4. The animal groups on AE+HFHF had less food intake overall compared to all other groups. would the animal housing or temp control have contributed to the decrease in blood glucose levels seen in the GTT test?
  5. It seems that the AE 500 HFHF group had worsened metabolic parameters, was this group observed to be more sedentary than other groups?
  6. As the liver histological results show that more hepatocyte ballooning was evident, did authors measure ALT levels in the blood?
  7. I recommend adding in the methods section (under the OGTT test) what blood was used for the test (eg tail bleed, etc)
  8. I noticed that the fasting blood glucose after treatment in figure 2C and figure 4A do not match. Can you please check.

Round 2

Reviewer 2 Report

Thank you for addressing the issues raised.